# Treatment of Metastatic Melanoma at First Diagnosis: Review of the Literature

**DOI:** 10.3390/life12091302

**Published:** 2022-08-24

**Authors:** Miguel-Angel Berciano-Guerrero, Mora Guardamagna, Elisabeth Perez-Ruiz, Jose-Miguel Jurado, Isabel Barragán, Antonio Rueda-Dominguez

**Affiliations:** 1Medical Oncology Intercenter Unit (Group of Translational Research in Cancer Immunotherapy), Regional and Virgen de la Victoria University Hospitals, Instituto de Investigación Biomédica de Málaga y Plataforma en Nanomedicina-IBIMA Plataforma BIONAND, 29010 Málaga, Spain or; 2Medical Oncology Intercenter Unit, Regional and Virgen de la Victoria University Hospitals, Instituto de Investigación Biomédica de Málaga y Plataforma en Nanomedicina-IBIMA Plataforma BIONAND, 29010 Málaga, Spain; 3Department of Medicine and Dermatology, Medical School University of Málaga, Campus Teatinos, Blvr. Louis Pasteur, 32, 29010 Málaga, Spain; 4Group of Pharmacoepigenetics, Department of Physiology and Pharmacology, Karolinska Institutet, 171 77 Stockholm, Sweden

**Keywords:** metastatic melanoma, first diagnosis, MUP

## Abstract

Metastatic melanoma (MM) is a pathological entity with a very poor prognosis that, until a few decades ago, had a low response rate to systemic treatments. Fortunately, in the last few years, new therapies for metastatic melanoma have emerged. Currently, targeted therapy and immunotherapy are the mainstays of the therapeutic arsenal available for patients with unresectable or metastatic melanoma. However, both clinical evolution and drug efficacy in melanoma patients are very different depending on the stage at which it is diagnosed. In fact, the aggressiveness of melanoma is different depending on whether it debuts directly as metastatic disease or if what occurs is a relapse after a first diagnosis at an early stage, although the biological determinants are largely unknown. Another key aspect in the clinical management of metastatic melanoma at first diagnosis strives in the different prognosis of melanoma of unknown primary (MUP) compared to melanoma of known primary (MPK). Understanding the mechanisms behind this, and the repercussion of implementing targeted and immune therapies in this specific form is crucial for designing diagnosis and treatment decision algorithms that optimize the current strategies. In this review article, we recapitulate the information available thus far regarding the epidemiology and response to immunotherapy treatments or targeted therapy in patients diagnosed with metastatic melanoma as a first diagnosis, with especial emphasis on the emerging specific information of the subpopulation formed by MUP patients.

## 1. Introduction

Despite representing 1% of skin tumors, melanoma represents 80% of skin cancer deaths [1]. Nevertheless, 90% of melanoma patients are initially diagnosed as stages I and II [2], with diagnosis of more advanced stages at a far lower incidence. Campaigns for primary and secondary prevention of melanoma have progressively decreased the incidence of thin melanoma [3].

Previously, many efforts have been made to understand patterns of disease progression, providing information on risks that have helped improve monitoring and the efficacy of new adjuvant treatments [4,5,6,7,8]. More recently, additional studies are aiming to improve not only survival in the initial stages, with better diagnosis and adjuvant therapies that are more effective, but also the treatment of unresectable or metastatic disease, regardless of the initial stage at diagnosis. The arrival of immune checkpoint inhibitors (ICI) and targeted therapy (TT) with BRAF inhibitors/MEK inhibitors (BRAFi/MEKi) has triggered a paradigm shift, facilitating a better understanding of the molecular biology of these tumors. Melanoma is, to date, the most immunogenic tumor [9]. The high rate of neoantigens allows the tumor cells to respond better to therapies that act on the immune system and the tumor microenvironment (TME). Moreover, the immunological perturbation induced by ICI could influence the response to other subsequent therapies, suggested by better outcomes of TT after ICI [10]. However, more studies are needed to confirm this strategy.

In the following review article, we have reviewed the epidemiology and molecular biology of MM as an initial diagnosis and have analyzed how it is currently managed in clinical trials and in the real world. This presentation of melanoma could have a special importance for the impact of current treatments. Therefore, we believe that this study holds particular utility for the delineation of effective therapeutic approaches that can impact the prognosis of the melanoma patients identified at this stage.

## 2. Epidemiology of MM at First Diagnosis

Melanoma is a tumor with a low incidence rate, but high mortality. For 2022, the estimated number of new cases of melanoma in situ or invasive in the United States is 97,920 and 99,780, respectively, with an estimated number of deaths of 7650 patients. This indicates an annual decrease of 4%, due to recent improvements in treatments in both adjuvant and advanced disease treatments [11]. This trend can be observed in other countries in the world.

Survival rates according to initial disease stage have been extensively described in large cohorts of patients [12,13,14,15], and the loss of efficacy towards advanced stages is a consolidated event (Table 1).

As mentioned above, fortunately, most melanomas are diagnosed in the very early stages, providing a better prognosis for the disease, in global terms.

Interestingly, a recent study presented epidemiological data on the incidence of cutaneous melanoma in Belgium and the Netherlands, using national cancer registries and analyzing different aspects, including stage. Although the incidence of melanoma was higher in the Netherlands than in Belgium, Belgium reported the highest incidence rate of stage IV disease at first diagnosis. Furthermore, this study not only showed different epidemiological aspects of the incidence, but also reported different survival rates. This reflects the difficulty of extrapolating information in this subgroup of patients, since it even appears difficult to explain these different incidences for biological issues in two neighboring countries [18].

On the other hand, a form of presentation of MM at initial diagnosis that is of special interest is melanoma of unknown primary (MUP). Indeed, such diagnosis would be excluded in the following scenarios: (1) Patients who do not receive a complete physical examination (including anogenital mucosa and ophthalmological examination); (2) Patients with evidence of previous orbital enucleation; (3) Patients with surgical procedures without histological documentation; and (4) Patients with lymph node involvement and the presence of a scar in the area of skin drained by the lymphatic basin [19]. Although several hypotheses have been described to explain this phenomenon, the theory that has gained the most strength, especially with current data on cancer immunology, is the involvement of immunological mechanisms on the primary tumor that lead to tumor regression. The incidence of MUP is reported to be 2.5–4% in most articles [20,21,22], although some report up to 10.5% of patients with metastatic melanoma. However, these do not specify the percentage of patients based on stages [23]. According to these publications, the presence of tumor regression in metastatic sites and low nodal burden were associated with favorable outcomes [24,25]. Therefore, MUP seems to have a better prognosis than melanoma of known primary (MKP). The different types of presentation of metastatic melanoma are presented in Figure 1.

## 3. Molecular Characterization of MM at First Diagnosis

In recent years, melanoma has been extensively characterized on a molecular basis in several studies. In fact, the latest World Health Organization (WHO) classification establishes nine categories of different melanoma subtypes, depending on the different driver mutations presented [26]. However, in routine practice, the most common mutations that have an approved therapeutic target (BRAF/MEK inhibition) are BRAF mutations, with other mutations that exceptionally require different treatment (NTRK fusions, etc.). Currently, efforts are being made to identify markers of efficacy and resistance to targeted therapy in metastatic melanoma [27].

The molecular characterization of patients with metastasis at presentation is not clear, nor is that of patients that present with MUP [28]. A recent article based on the nationwide Flatiron Health electronic database describes the frequency of BRAF mutations and the use of different therapeutic strategies depending on several factors, such as the stage at initial diagnosis [17]. Of the 4459 patients, 1191 had been diagnosed with stage IV at diagnosis, representing 26.7% of the total number of patients analyzed. These patients were diagnosed at a mean age of 66 years, presenting a good general condition (ECOG 0–1) and patterns of use of different systemic treatments similar to the general population (see Table 1). The only difference reported in this study is more frequent treatment with BRAF inhibitors in those patients who had a BRAF mutation and were initially diagnosed as stage IV, although these differences may be due to the high rate of patients excluded from the study. Additionally, the different patterns of use of treatments depending on the year can explain these differences.

Therefore, the molecular characterization of patients with metastasis at the first diagnosis is not clearly defined. With these data, it would be difficult to explain whether patients presenting with stage IV at initial diagnosis have a cancer biology that is different from those diagnosed in earlier stages.

## 4. Representation of MM at First Diagnosis in Pivotal Studies

To search for epidemiological data on how patients respond to different treatments depending on the initial stage at diagnosis, we reviewed published clinical trial information, as well as complementary material. The systematic search methodology is shown in the Appendix A.

### 4.1. Targeted Therapy

Currently, there are several studies that establish BRAF-targeted therapy as standard treatment for patients with metastatic melanoma who carry a BRAF mutation. The combinations vemurafenib-cobimetinib, dabrafenib-trametinib and encorafenib-binimetinib constitute the first line of treatment for patients with BRAF-mutated metastatic melanoma. The main characteristics of these studies are shown in Table 2.

The experience and knowledge accumulated over the last few years have elucidated that BRAF-mutated melanoma is a more aggressive tumor than BRAF wild-type melanoma, with greater growth rate and a greater capacity for metastasizing [36]. Therefore, initial molecular characterization may also contribute to predict such behavior and how targeted therapy efficacy is affected by that. Importantly, this information is so far lacking in the pivotal studies of this therapeutic approach.

Furthermore, new strategies with targeted therapy, such as triplet treatments that include targeted therapy and immunotherapy, may provide data on the clinical evolution and response to treatment in these MM patients. However, data on this subpopulation are thus far not available in these combination strategies trials such as the COMBI-i study, which combines spartalizumab with dabrafenib and trametinib, or the IMspire150 study that combines atezolizumab with vemurafenib and cobimetinib [37,38].

Similarly, MUP patients were included in the targeted therapy pivotal studies, although their outcomes were not reported. Therefore, the efficacy of these treatments in this subgroup of patients remains to be specified (Table 2).

### 4.2. Immunotherapy

The data from the pivotal studies with approved immunotherapeutic drugs that constitute the first line of treatment for metastatic melanoma are presented in Table 3.

Information on patients diagnosed with stage IV at debut is not reported in any of the studies. Other similar clinical trials and their updates did not specify this subpopulation in their study population either [16,46,47,48,49].

Theoretically, it seems reasonable that immunotherapy may play an essential role in for the treatment of patients with MUP, given the implication of immunological mechanisms. However, there is no information regarding the MUP population reported in these studies.

In addition, therapeutic strategies that combine immunotherapy with several other drugs are currently being developed. However, none of the new combinations such as pembrolizumab with lenvatinib in the LEAP-004 study [50], bempegleukin and nivolumab in the PIVOT-02 study [51] or, more recently, nivolumab and relatlimab in the RELATIVITY-047 study [52] reported data on this subgroup of patients.

Similar to the targeted therapy studies, MUP patients were included in the trials with approved immunotherapy, but specific outcomes were not shown (Table 3).

### 4.3. Chemotherapy

Although it features in later lines of treatment, chemotherapy continues to be a therapeutic weapon for the treatment of metastatic melanoma. However, the development of this therapeutic strategy has been relegated to a less important priority. In the latest clinical trials reported using chemotherapy-based schemes, the population was described based on the initial stage at diagnosis, so we do not know whether this population would respond to these treatments, based on that variable [48,53,54].

## 5. Representation of MM at First Diagnosis in Real-World Data

Since clinical trials did not report data on which we could draw conclusions, our group investigated real-world data.

New York University (NYU) recently published a study on their experience with the new systemic treatments for MM. In this article, Utter et al. show the data of patients treated with ICI or TT for MUP at their institution, comparing them with the patients of this subpopulation reported in the literature. Both subgroups were compared with reported MKP patients. MUP patients treated at NYU had better outcomes with ICI but worse with TT than the reported MUP patients in the literature. However, NYU MUP patients had worse outcomes than MKP patients in clinical trials. In the ICI subgroup, most patients were treated with anti-CTLA-4 [55], which could explain the worse results than those found in the literature.

Subsequently, a German group published their experience with MUP patients treated with anti-PD1 [56]. Although the number of patients was even lower (nine patients), the study concludes that patients with MUP have a lower risk of developing melanoma-specific death (MSD).

A retrospective study was also recently published in which the particularities of MUP patients of an Italian center are described. This study analyzed the characteristics and the different therapeutic modalities established in 127 patients, comparing them with MPK patients. No differences were found between the survival rates of patients treated with immunotherapy or with targeted therapy [57]. Despite being the study with the largest number of patients, the authors highlight the heterogeneity of the population and the therapeutic strategies received, so further studies are needed to confirm their conclusions.

There are many retrospective real-life studies on the efficacy and safety of drugs in patients diagnosed with unresectable stage III or stage IV melanoma. However, with the initial diagnosis of stage IV at onset, it is not so frequent, and it is necessary to investigate the analysis of subpopulations in previous studies. In this sense, in 2012, a multicenter longitudinal retrospective study of several European countries—the MELODY study—was presented on treatment patterns and outcomes among patients in this population [58,59]. This study collected information from 776 patients, of whom 104 (14%) were unresectable stage III or stage IV at first diagnosis.

This study, which is prior to the establishment of immunotherapy and targeted therapy, reports information on OS in patients with a diagnosis of advanced disease at any time but does not report more information on patients diagnosed at their first diagnosis.

In 2016, a prospective and retrospective observational study was reported on the epidemiology of patients with metastatic melanoma in the province of Ontario (Canada) [60]. In this study, the characteristics of the patients were described, based on the initial stage at diagnosis. The patients initially diagnosed with stage IV are shown in Table 1. Survival at the end of the study was 79.9, 71.0 and 52% in those initially diagnosed as stage 0–II, III and IV, respectively. This study shows that initial diagnosis at an earlier stage has a better prognosis than those diagnosed directly at more advanced stages. Moreover, a Cox proportional hazards model identified initial stage as a significant predictor of 1- and 2-year survival. The authors of this study are already aware of the temporality bias, given the different diagnostic options (fundamentally BRAF mutation determination) and therapeutic options with respect to the current standard.

A recent study, reported by Zhou et al. on data from the Netherlands Cancer Registry (NCR), provides the data that 60% of deaths from metastatic melanoma are due to debut with extended disease. In this study, the characteristics of patients with early stages that progress to metastatic melanoma are analyzed, although they do not provide data on those initially diagnosed with stage IV melanoma, except that they constitute 13.8% of all stages IV (223/1613) [61].

Finally, in ESMO 2021, data from a registry of the Spanish Melanoma Group (GEM) were reported. However, only patient data at the time of study inclusion were shown [62].

## 6. Limitations

Our review article highlights the absence of publications on this advanced disease subpopulation at first diagnosis in all of the pivotal studies that led to the achievement of approval of current drugs for patients with metastatic melanoma. We believe that the scant evidence shown could be biasing therapeutic decisions about the treatment of unresectable or metastatic melanoma. Most of the identified studies show bias in terms of the fact that they correspond to a period where diagnosis and treatment do not correspond to current management.

## 7. Conclusions

Currently, the biological and clinical aspects that may occur between metastatic melanoma diagnosed at first diagnosis or as an early stage recurrence are unknown. MM at initial diagnosis is a more aggressive entity than when melanoma is initially resected at an early stage and recurs over time. In addition, we know that MUP is a form of presentation of MM at first diagnosis, with better prognosis than MKP. Despite being an entity with a different prognosis, no information is available on the efficacy of targeted therapy and immunotherapy treatments. In most cases, the time of the diagnosis of stage IV is not specified, so more data are needed to shed light on this clinical situation that we find in daily medical practice.

## Figures and Tables

**Figure 1 life-12-01302-f001:**
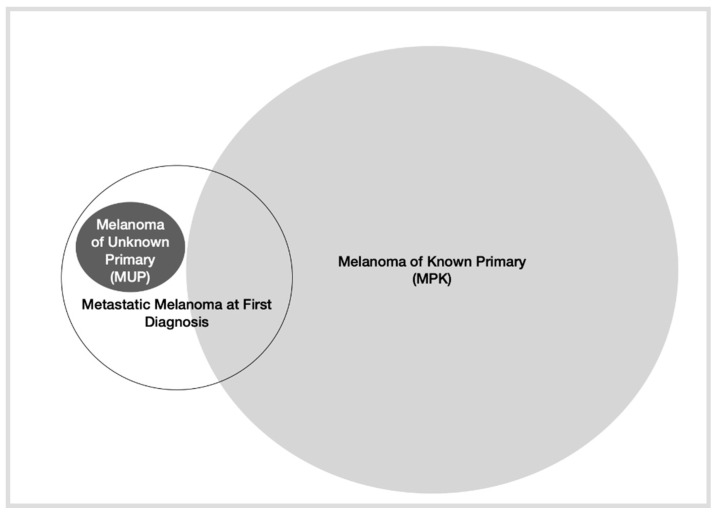
Depiction of different types of presentation of metastatic melanoma.

**Table 1 life-12-01302-t001:** Studies reporting data on metastatic melanoma at initial diagnosis.

Article	Ernst et al. [16]		Hill et al. [17]	
Staging ^a^	Stage IV at Initial Diagnosis	All	Stage IV at Initial Diagnosis	All
**All patients (n%)**	78 (9.6)	810 (100)	1191 (26.7)	4459
**Stage**	Stage IV	Stage 0-IV	Stage IV	III-IV/Recurrence
**Mean age (years)**	60.69	58.74	66	64.4
**Sex [n (%)]**				
**Men**	45 (9.3)	485 (59.9)	-	-
**Women**	33 (10.2)	325(40.1)	-	-
**ECOG (%)**				
**0–1**			665 (55.8)	2619 (58.7)
**2**			96 (8.1)	248 (5.5)
**3–4**			44 (3.7)	91 (2.0)
**Mean time to recurrence (years)**	3.84 ± 5.19	4.94 ± 6.69		
**Mutation type [n (%)]**				
**BRAF**	11 (21.6)	51 (6.3)	444 (37.3)	1314 (42)
**CKIT**	1 (16.7)	6 (0.7)	-	-
**NRAS**	0	1 (0.1)	-	-
**MEK**	0	0	-	-
**GNAO**	0	0	-	-
**GNA11**	0	0	-	-
**Metastatic [n (%)]**				
**Yes**	11 (21.6)	354 (44.0)		
**No**	1 (16.7)	454 (56.1)		
**Resectable [n (%)]**				
**Yes**	0	704 (86.9)		
**No**	78 (73.6)	106 (13.1)		
**Developed metastatic during course of disease**				
**No**	78 (100)	454 (56.0)		
**Yes**	0	346 (42.7)		
**Median Overall Survival [months (95% CI)]**				
**Initially stage 0–II**	111.3			
**Initially stage III**	(95.6–131)			
**Initially stage IV**	76.3 (59.3–93.3)			
	59.9 (38.2–81.7)			
**First Line Treatment (%)**				
**Immunotherapy**			665 (73.1)	1652 (73.5)
**BRAF inhibitor**			186 (20.4)	393 (17.5)
**Clinical trial**			28 (3.1)	106 (4.7)
**Chemotherapy**			19 (2.1)	51 (2.3)
**Interferon**			5 (0.5)	14 (0.6)
**IL-2**			1 (0.1)	4 (0.2)

^a^ TNM Classification according to the 7th edition.

**Table 2 life-12-01302-t002:** Main pivotal studies of currently approved targeted therapy for metastatic melanoma.

Clinical Trial	COBRIM [29,30]	COMBI-V/D [31,32,33]	COLUMBUS [34,35]
**Experimental Drug**	Cobi + Vem	Dabra-Trame	Enco-Bini
**Control Drug**	Pbo + Vem	Vemu/Dabra	Vemu
**Stage at randomization ^a,b^ (%)**			
**III**	9	2	4.7
**IV (M1c)**	91	98	95.3
**Stage IV at first diagnosis ^a,b^**	NR	NR	NR
**mPFS ^a^**	12.6	11.1	14.9
**mOS ^a^**	22.5	25.9	33.6
**mOS by stage ^a,b^**: **III** **M1a** **M1b** **M1c**	NR 54.8 19.4 18.9	NR HR III-M1b vs. M1c 0.76 (0.58–1)	NR
**MUP patients**			
**Included**	Yes	Yes	Yes
**Outcomes**	NR	NR	NR

^a^ In experimental arm; ^b^ TNM Classification according to7th edition; NR: Not reported; Cobi: cobimetinib, Vem: vemurafenib, Pbo: placebo, Dabra: dabrafenib, Trame: trametinib, Enco: encorafenib, Bini: Binimetinib.

**Table 3 life-12-01302-t003:** Main pivotal studies of currently approved immunotherapy for metastatic melanoma.

Clinical Trial	CHECKMATE-066 [39,40,41]	KEYNOTE-006 [42,43]	CHECKMATE-067 [44,45]
**Experimental Drug**	Nivo	Pem	Nivo-Ipi
**Control Drug**	DTIC	Ipi	Ipi/Nivo
**Stage at randomization ^a,b^ (%)**			
**III**	NR	3.1	
**IV (M1c)**	61% M1c	96.9 (66.2)	(58.9)
**Stage IV at first diagnosis ^a,b^**	NR ^c^	NR	NR
**mPFS ^a^**	5.1	8.4	11.5
**mOS ^a^**	37.3	31.1	60.0
**mOS by stage ^a,b^**: **III** **M1a** **M1b** **M1c**	NR	NR	NR
**MUP patients**			
**Included**	Yes	Yes	Yes
**Outcomes**	NR	NR	NR

^a^ In experimental arm; ^b^ TNM Classification according to 7th edition; ^c^ At least 15.2% received adjuvant treatment; NR: Not reported; Nivo: nivolumab, DTIC: dacarbazine, Pem: pembrolizumab, Ipi: ipilimumab.

## Data Availability

Not applicable.

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
