# Peer review of "Treatment of Metastatic Melanoma at First Diagnosis: Review of the Literature"

_life, 2022, doi:10.3390/life12091302_

Round 1

Reviewer 1 Report

The review article by Berciano-Guerrero and colleagues highlights the paucity of data surrounding the treatment and outcomes of patients first diagnosed with disseminated or metastatic melanoma.  The review covers the lack of data contained in publications of clinical trials, as well as that obtained from real world clinical experience.  The authors highlight the need for further examination of the published data, as well as ongoing studies to determine the best therapeutic avenue for this subset of melanoma patients.

The manuscript referenced all of the pivotal clinical trial and was comprehensive.  The only major comment was regarding the section beginning at line 181.  I feel that a brief description of the outcome of the patients referred to here would be appropriate.

The manuscript would benefit from some careful editing.  In places, the language didn’t correctly convey the message that the authors were trying to get across to the reader (e.g. the use of double negative terms).  Please find below some suggestions to clarify the content for the reader.

Line 30: “being the diagnosis of more advanced stages a fact of lower incidence.” could be changed to “with diagnosis of more advanced stages at a far lower incidence.”

Line 34: “Classically” could be changed to “Previously”.

Line 39: “adjuvant therapies more effective, but” could be changed to “adjuvant therapies that are more effective, but”.

Line 42: “Melanoma is, to date, the most immunogenic tumor.”  A reference following this statement would be appropriate.

Line 50: “biology of MM as initial diagnosis” could be changed to “biology of MM as an initial diagnosis”.

Line 51: “clinical trials and in Real World” could be changed to “clinical trials and in the real world” or “clinical trials and in real world situations”.

Lines 84-86: “The incidence of MUP is 2.5-4% in some reported articles [17-19], although some reach 10.5% of patients with metastatic melanoma, although it does not specify the percentage of patients based on stages [20].” could be changed to “The incidence of MUP is reported to be 2.5-4% in most articles [17-19], although some reports up to 10.5% of patients with metastatic melanoma; however these do not specify the percentage of patients based on stages [20].”

Line 94: “…is not clear. Neither of those patients who presented as MUP” could be changed to “is not clear, nor is that of those patients that present with MUP”.

Line 115: “initial stage at diagnosis, we found it interesting to review the published” could be changed to “initial stage at diagnosis, we reviewed the published”.

Line 134: “Nor have the new strategies with targeted therapy..” could be changed to “In addition / Additionally / Furthermore have the new strategies with targeted therapy..”.

Lines 153-155: “In this sense, it seems reasonable that immunotherapy plays an essential role in patients with MUP, given the implication of immunological mechanisms. Neither is the MUP population reported in these studies.” could be changed to “In this sense, it seems reasonable that immunotherapy may play an essential role in therapy for patients with MUP, given the implication of immunological mechanisms. However, there is no information regarding the MUP population reported in these studies.”

Line 158: “…currently being developed. Nor have the new combinations…” could be changed to “…currently being developed. However, none of the new combinations…”

Line 237: “…at first diagnosis in none of the pivotal studies that achieved…” could be changed to “…at first diagnosis in all of the pivotal studies that achieved…”

Author Response

REVIEWER 1:

The review article by Berciano-Guerrero and colleagues highlights the paucity of data surrounding the treatment and outcomes of patients first diagnosed with disseminated or metastatic melanoma.  The review covers the lack of data contained in publications of clinical trials, as well as that obtained from real world clinical experience.  The authors highlight the need for further examination of the published data, as well as ongoing studies to determine the best therapeutic avenue for this subset of melanoma patients.

Thanks to the reviewer for this comment. We believe that this article can be a good way to bring to light a need in clinical practice. We appreciate all kinds of comments to improve the formal and content aspects of the manuscript.

The manuscript referenced all of the pivotal clinical trial and was comprehensive.  The only major comment was regarding the section beginning at line 181.  I feel that a brief description of the outcome of the patients referred to here would be appropriate.

Thanks to the reviewer for pointing this issue as this would help improving the manuscript. We have described this article more in the new version.

The manuscript would benefit from some careful editing.  In places, the language didn’t correctly convey the message that the authors were trying to get across to the reader (e.g. the use of double negative terms).  Please find below some suggestions to clarify the content for the reader.

Manuscript has undergone English language editing by MDPI.

Line 30: “being the diagnosis of more advanced stages a fact of lower incidence.” could be changed to “with diagnosis of more advanced stages at a far lower incidence.”

Changed.

Line 34: “Classically” could be changed to “Previously”.

Changed

Line 39: “adjuvant therapies more effective, but” could be changed to “adjuvant therapies that are more effective, but”.

Changed

Line 42: “Melanoma is, to date, the most immunogenic tumor.”  A reference following this statement would be appropriate.

Lawrence MS, Stojanov P, Polak P, et al. Mutational heterogeneity in cancer and the search for new cancer-associated genes. Nature. 2013;499(7457):214-218. doi:10.1038/nature12213

Line 50: “biology of MM as initial diagnosis” could be changed to “biology of MM as an initial diagnosis”.

Changed

Line 51: “clinical trials and in Real World” could be changed to “clinical trials and in the real world” or “clinical trials and in real world situations”.

Changed

Lines 84-86: “The incidence of MUP is 2.5-4% in some reported articles [17-19], although some reach 10.5% of patients with metastatic melanoma, although it does not specify the percentage of patients based on stages [20].” could be changed to “The incidence of MUP is reported to be 2.5-4% in most articles [17-19], although some reports up to 10.5% of patients with metastatic melanoma; however these do not specify the percentage of patients based on stages [20].”

Changed

Line 94: “…is not clear. Neither of those patients who presented as MUP” could be changed to “is not clear, nor is that of those patients that present with MUP”.

Changed

Line 115: “initial stage at diagnosis, we found it interesting to review the published” could be changed to “initial stage at diagnosis, we reviewed the published”.

Changed

Line 134: “Nor have the new strategies with targeted therapy..” could be changed to “In addition / Additionally / Furthermore have the new strategies with targeted therapy..”.

Changed

Lines 153-155: “In this sense, it seems reasonable that immunotherapy plays an essential role in patients with MUP, given the implication of immunological mechanisms. Neither is the MUP population reported in these studies.” could be changed to “In this sense, it seems reasonable that immunotherapy may play an essential role in therapy for patients with MUP, given the implication of immunological mechanisms. However, there is no information regarding the MUP population reported in these studies.”

Changed

Line 158: “…currently being developed. Nor have the new combinations…” could be changed to “…currently being developed. However, none of the new combinations…”

Changed

Line 237: “…at first diagnosis in none of the pivotal studies that achieved…” could be changed to “…at first diagnosis in all of the pivotal studies that achieved…”

Changed

Reviewer 2 Report

The authors through this review had the intention to clarify the possible diagnostic approach to treat melanoma depending on the stage and clinical management at initial diagnosis by referring to epidemiological studies, melanoma molecular biology and by clinical studies on existing therapies (immunotherapy or targeted therapies).

Major point to develop:

- it is necessary to re-evaluate the edition, the English and the general plagiarism of the paperr

- Abstract: The points made in the abstract do not correspond to the points made in the review. There is a lack of clarity on why to study unknown primary metastatic melanoma (MUP) which appears in the text as a whole but not in the abstract. The summary should be reformulated to emphasize the importance of characterization of MUPs, their molecular identification from an epidemiological point of view, and the importance of the study of MUPs.

- line 28: the comparison of the different skin cancers does not use the right reference, a more appropriate reference is needed: "Despite representing ..cancer deaths ". 

- line 40: for a clearer definition of what iBRAF/MEK means

- line 42-44: a bibliographic reference would be helpful to expand on your statement "Melanoma ... tumor microenvironment (TME). "

- line 51: a more explicit explanation of the term in Real World" footprint in the Hill et al 2022 article, but in this case that you are developing it does not fit

- line 59: you identify the United States as the reference for the incidence of melanoma, whereas the previous studies (table 1) were done in Canada (Ernet et al 2016), I suggest that you explain that this trend can be observed in other countries around the world.

- line 62: the web link is not current, please refer to a web link more in line with your comments.

- line 70-77: the use of "two neighboring countries" is not well understood at the beginning of the paragraph It is not clear at the beginning of the paragraph why "two neighboring countries" is used, and it is explained later that these are the Netherlands and Belgium. Rewrite this paragraph for better readability and understanding of the difficulty and misunderstanding of incidence values in melanoma - link with the article Reyn B et al 2021 (ref 15).

- line 80: explained in more detail the origin of the term "Melanoma of Unknown Primary (MUP): the diagnostic inclusion criteria how to distinguish them from known primary melanoma (MKP) for a more accurate understanding instead of "It is an entity described more than years ago" ref (16).It should be placed as a major player in your investigation 

-line 187-190: it is preferable in a review to feed with references that are effective and underlined the importance of an axis of research that is still poorly studied. Thus "Although ... In this sense, our group is developing a study in collaboration with other centers for a better ...response to treatment of patients with MUP (pending publication), makes no sense in a review if your work is not yet published.

-line 91-111: part 3 "Molecular characterization of MM at first diagnosis" is poorly listed in terms of molecular characterization and refers to only one reference, that of HIll et al 2022, whereas numerous studies have been focused on molecularly elucidating the heterogeneity that exists in primary melanomas that can explain different outcomes of the disease progression This section needs to be more detailed from a molecular point of view. 

- line 127 reference 25 which is a review does not highlight the distinct role of BRAF vs BRAFWT mutation in the ability to metastasize "BRAF mutated melanoma is a more aggressive... . Probably BRAFV600E and BRAFWT melanoma have two distinct genetic mechanisms associated with disease progression. I suggest getting closer to an experimental approach to support your claims.

The article in this state appears uncomplete with numerous ambiguities in substance and form. It requires much explanations and explicit meanings to understand better your scientific approach and the importance of conducting these epidemiological studies

Author Response

REVIEWER 2:

The authors through this review had the intention to clarify the possible diagnostic approach to treat melanoma depending on the stage and clinical management at initial diagnosis by referring to epidemiological studies, melanoma molecular biology and by clinical studies on existing therapies (immunotherapy or targeted therapies).

Major point to develop:

  • it is necessary to re-evaluate the edition, the English and the general plagiarism of the paperr

Manuscript has undergone English language editing by MDPI.

  • Abstract: The points made in the abstract do not correspond to the points made in the review. There is a lack of clarity on why to study unknown primary metastatic melanoma (MUP) which appears in the text as a whole but not in the abstract. The summary should be reformulated to emphasize the importance of characterization of MUPs, their molecular identification from an epidemiological point of view, and the importance of the study of MUPs.

Changed

  • line 28: the comparison of the different skin cancers does not use the right reference, a more appropriate reference is needed: "Despite representing ..cancer deaths ". 

 Changed

  • line 40: for a clearer definition of what iBRAF/MEK means

Changed

  • line 42-44: a bibliographic reference would be helpful to expand on your statement "Melanoma ... tumor microenvironment (TME). “

Lawrence MS, Stojanov P, Polak P, et al. Mutational heterogeneity in cancer and the search for new cancer-associated genes. Nature. 2013;499(7457):214-218. doi:10.1038/nature12213

  • line 51: a more explicit explanation of the term in Real World" footprint in the Hill et al 2022 article, but in this case that you are developing it does not fit

Changed

  • line 59: you identify the United States as the reference for the incidence of melanoma, whereas the previous studies (table 1) were done in Canada (Ernet et al 2016), I suggest that you explain that this trend can be observed in other countries around the world.

Changed

  • line 62: the web link is not current, please refer to a web link more in line with your comments.

Changed

  • line 70-77: the use of "two neighboring countries" is not well understood at the beginning of the paragraph It is not clear at the beginning of the paragraph why "two neighboring countries" is used, and it is explained later that these are the Netherlands and Belgium. Rewrite this paragraph for better readability and understanding of the difficulty and misunderstanding of incidence values in melanoma - link with the article Reyn B et al 2021 (ref 15).

Changed

  • line 80: explained in more detail the origin of the term "Melanoma of Unknown Primary (MUP): the diagnostic inclusion criteria how to distinguish them from known primary melanoma (MKP) for a more accurate understanding instead of "It is an entity described more than years ago" ref (16).It should be placed as a major player in your investigation 

Changed

-line 187-190: it is preferable in a review to feed with references that are effective and underlined the importance of an axis of research that is still poorly studied. Thus "Although ... In this sense, our group is developing a study in collaboration with other centers for a better ...response to treatment of patients with MUP (pending publication), makes no sense in a review if your work is not yet published.

Changed

-line 91-111: part 3 "Molecular characterization of MM at first diagnosis" is poorly listed in terms of molecular characterization and refers to only one reference, that of HIll et al 2022, whereas numerous studies have been focused on molecularly elucidating the heterogeneity that exists in primary melanomas that can explain different outcomes of the disease progression This section needs to be more detailed from a molecular point of view. 

Changed

  • line 127 reference 25 which is a review does not highlight the distinct role of BRAF vs BRAFWT mutation in the ability to metastasize "BRAF mutated melanoma is a more aggressive... . Probably BRAFV600E and BRAFWT melanoma have two distinct genetic mechanisms associated with disease progression. I suggest getting closer to an experimental approach to support your claims.

We have changed the reference for one of the original studies of BRAF:

Davies H, Bignell GR, Cox C, et al. Mutations of the BRAF gene in human cancer. Nature. 2002;417(6892):949-954. doi:10.1038/nature00766

The article in this state appears uncomplete with numerous ambiguities in substance and form. It requires much explanations and explicit meanings to understand better your scientific approach and the importance of conducting these epidemiological studies

We appreciate the effort and thoughtful insights on the part of the reviewer. We believe that they strengthen the evidence and robustness of the article. Many expressions have been changed, concepts have been defined, new references have been added and the language has been revised, which we believe have substantially improved the manuscript. We hope the article is ready to publish.

Round 2

Reviewer 2 Report

1- There has been an effort to revise the paper, however, the abstract does not seem sufficiently detailed and lacks the major points about the central role of MUPs in the clinical evaluation of patients.

2- Cette correction n est pas suffisante avec de nombreuses répétions syntaxique. une partie entière doit être mis en évidence sur le mélanome d'origine inconnue (MUP).On the other hand, a form of presentation of MM at initial diagnosis that is of special interest is melanoma of unknown primary (MUP). A definitive diagnosis of MUP occurs in the absence of a primary cutaneous, ocular, or mucosal melanoma after a thorough physical examination. Patients who do not receive a complete physical examination (including anogenital mucosa and ophthalmological examination); those with evidence of previous orbital enucleation; those with surgical procedures without histological documentation; and those with lymph node involvement and the presence of a scar in the area of skin drained by the lymphatic basin are excluded from this diagnosis [17]. 

3- Changing the reference is not sufficient but does detail the importance of the role of BRAFV600E vs BRAFwt in the ability to metastasize.Since pathological evidence in patients shows that BRAFwt melanomas do metastasize , therefore, the idea is not to confront these two entities on their ability to metastasize.  The experience and knowledge accumulated over the last few years have elucidated that BRAF-mutated melanoma is a more aggressive tumor than BRAF wild-type melanoma, with greater growth and a greater capacity for metastasizing [28]. Therefore, it may be important to identify patients who were initially diagnosed at stage IV and to see the efficacy of TT in this subgroup, given that the molecular and clinical characteristics are unknown. However, none of the pivotal studies reported data from such patients within their analysis.

Author Response

REVIEWER 2- ROUND 2

1- There has been an effort to revise the paper, however, the abstract does not seem sufficiently detailed and lacks the major points about the central role of MUPs in the clinical evaluation of patients.

Thanks to the reviewer for insisting on improving the abstract, since we consider that its quality has been improved with his comments. The abstract has been changed in an attempt to reinforce the meaning of the review, as well as emphasizing the need for reviewing MUP, in particular given its importance when defining better diagnostic and therapeutic strategies. 

2- Cette correction n est pas suffisante avec de nombreuses répétions syntaxique. une partie entière doit être mis en évidence sur le mélanome d'origine inconnue (MUP).On the other hand, a form of presentation of MM at initial diagnosis that is of special interest is melanoma of unknown primary (MUP). A definitive diagnosis of MUP occurs in the absence of a primary cutaneous, ocular, or mucosal melanoma after a thorough physical examination. Patients who do not receive a complete physical examination (including anogenital mucosa and ophthalmological examination); those with evidence of previous orbital enucleation; those with surgical procedures without histological documentation; and those with lymph node involvement and the presence of a scar in the area of skin drained by the lymphatic basin are excluded from this diagnosis [17]. 

This part refers to a previous review by the same reviewer asking to define the specific inclusion criteria for the diagnosis of MUP. In the current version of the manuscript, we have revised the syntactic structure of the paragraph to make it clearer to the reader.

3- Changing the reference is not sufficient but does detail the importance of the role of BRAFV600E vs BRAFwt in the ability to metastasize.Since pathological evidence in patients shows that BRAFwt melanomas do metastasize , therefore, the idea is not to confront these two entities on their ability to metastasize.  The experience and knowledge accumulated over the last few years have elucidated that BRAF-mutated melanoma is a more aggressive tumor than BRAF wild-type melanoma, with greater growth and a greater capacity for metastasizing [28]. Therefore, it may be important to identify patients who were initially diagnosed at stage IV and to see the efficacy of TT in this subgroup, given that the molecular and clinical characteristics are unknown. However, none of the pivotal studies reported data from such patients within their analysis.

We appreciate the reviewer comment. With this paragraph, we aimed at strengthening the idea that specific mutations increase the likelihood of the melanoma to be metastatic at first diagnosis and therefore initial molecular characterization is even more important. We also want to remark that so far no pivotal study on the matter has specifically evaluated the particular clinical features and impact of targeted therapy in BRAF mutated patients with melanoma at first diagnosis.

Round 3

Reviewer 2 Report

no further feedback